# Hyperbolic optics and superlensing in room-temperature KTN from self-induced k-space topological transitions

Yehonatan Gelkop[1], Fabrizio Di Mei [2], Sagi Frishman[1], Yehudit Garcia[1,3], Ludovica Falsi[2,4], Galina Perepelitsa[1], Claudio Conti [2,5], Eugenio DelRe [2,5 ✉] & Aharon J. Agranat[1,3]

A hyperbolic medium will transfer super-resolved optical waveforms with no distortion, support negative refraction, superlensing, and harbor nontrivial topological photonic phases. Evidence of hyperbolic effects is found in periodic and resonant systems for weakly diffracting beams, in metasurfaces, and even naturally in layered systems. At present, an actual hyperbolic propagation requires the use of metamaterials, a solution that is accompanied by constraints on wavelength, geometry, and considerable losses. We show how nonlinearity can transform a bulk KTN perovskite into a broadband 3D hyperbolic substance for visible light, manifesting negative refraction and superlensing at room-temperature. The phenomenon is a consequence of giant electro-optic response to the electric field generated by the thermal diffusion of photogenerated charges. Results open new scenarios in the exploration of enhanced light-matter interaction and in the design of broadband photonic devices.

[1] The Department of Applied Physics, The Hebrew University, Jerusalem 9190401, Israel. [2] Dipartimento di Fisica, Università di Roma "La Sapienza", 00185 Rome, Italy. [3] The Brojde Center for Innovative Engineering and Computer Science, The Hebrew University, Jerusalem 9190401, Israel. [4] Dipartimento S.B.A.I., Sezione di Fisica, Università di Roma "La Sapienza", 00161 Rome, Italy. [5] ISC-CNR, Università di Roma "La Sapienza", 00185 Rome, Italy. ✉email: eugenio.delre@uniroma1.it

Wave behavior is governed by the topology of the iso-frequency surface $\mathcal{M}_\mathbf{k}$, the k-space manifold of plane-wave solutions of angular frequency $\omega$ that obey the dispersion relation $\mathbf{k} = \mathbf{k}(\omega)$. While standard dielectrics have a closed-surface topology, $\mathcal{M}_\mathbf{k}$ in hyperbolic media is characterized by a fundamentally different open-surface topology (see k-Space Topology Section in Methods)[1–8]. Hyperbolic phenomenology can emerge in photonic crystals and waveguide arrays[9–11], metasurfaces[12,13], layered media[14–17], and metamaterials[18–21]. A hyperbolic medium placed along an optical circuit then auto-matically implies one or more k-space topological transitions for the transmitted waves[22–29], a photonic equivalent of Bloch-electron Lifshitz transitions[30–33]. As discussed in the Self-Induced Topology Section in Methods, the diffusive photorefractive non-linearity present in near-transition paraelectric ferroelectrics[34,35] with a giant electro-optic response[36–39] causes $\mathcal{M}_\mathbf{k}$ to become $(1 - \alpha^2)(k_x^2 + k_y^2) + k_z^2 = k_0^2 n^2$, where the dimensionless parameter $\alpha = L/\lambda$ is determined by the diffusive length scale $L = 4\pi n^2 \varepsilon_0 \sqrt{g} \chi_{PNR}(K_B T/q)$. Here $g$ is the quadratic electro-optic coefficient, $\chi_{PNR}$ the low-frequency susceptibility dominated by super-cooled polar-nanoregions (PNRs), and $K_B T/q$ is the thermal voltage. This implies a passage from a closed-surface topology for $\alpha < 1$ (see Fig. 1A, blue surfaces) to an open-surface two-sheet hyperbolic topology for $\alpha > 1$ (Fig. 1A, yellow surfaces) as $\alpha$ is swept through the $\alpha = 1$ critical value. The effect of a transition from $\alpha = 0$ to $\alpha > 1$ is illustrated in Fig. 1B for the simplified 1+1D case, i.e., when dynamics only depend on one transverse coordinate $x$

and the propagation coordinate $z$ (so that $(1 - \alpha^2)k_x^2 + k_z^2 = k_0^2 n^2$). As light passes from the elliptical medium (air) to the hyperbolic medium (the ferroelectric with $\alpha > 1$), the boundary conservation of the transverse component of $k_x = k'_x$ causes the transmitted wave to undergo negative refraction: the original Poynting vector $\mathbf{S}$ is redirected along the normal to the hyperbolic isofrequency surface, $\mathbf{S}'$, opposite to what normally occurs in standard refraction. $\mathbf{S}'$ forms an angle $\tan\theta \simeq \sqrt{\alpha^2 - 1}$ with the normal to the boundary surface for the part of the hyperbolic branch that can be approxi-mated by an asymptote (dashed line). As illustrated in Fig. 1C, for a localized beam composed of a spectrum of plane waves on the $\alpha = 0 \to \alpha > 1$ boundary (the 'object' point), the transition amounts to the splitting of the original beam into two beams, each inheriting only half of the input transverse spatial spectrum ($k_x > 0$ and $k_x < 0$). Insomuch that each half of the spectrum principally generates waves in the $\alpha > 1$ medium with a $\mathbf{k}'$ that lies on the asymptotic branches, each beam undergoes negligible diffraction and propa-gates forming and angle $\pm\theta$ to the normal of the boundary, crossing inside the crystal (the 'internal focus'). In detail (see Fig. 1C right), the transition implies the topologically non-trivial crossing of the $k_x > 0$ spectrum to the beam with negative x-component Poynting vector, while the negative spectrum beam $k_x < 0$ propagates with a positive x-component. The two crossing beams are topologically protected, that is, they cannot suffer further cascaded splitting in the x direction because they only occupy one of the two arms of the hyperbola corresponding to its half-spectrum. If the beams then suffer an inverse transition $\alpha > 1 \to \alpha = 0$ (exiting the sample), they

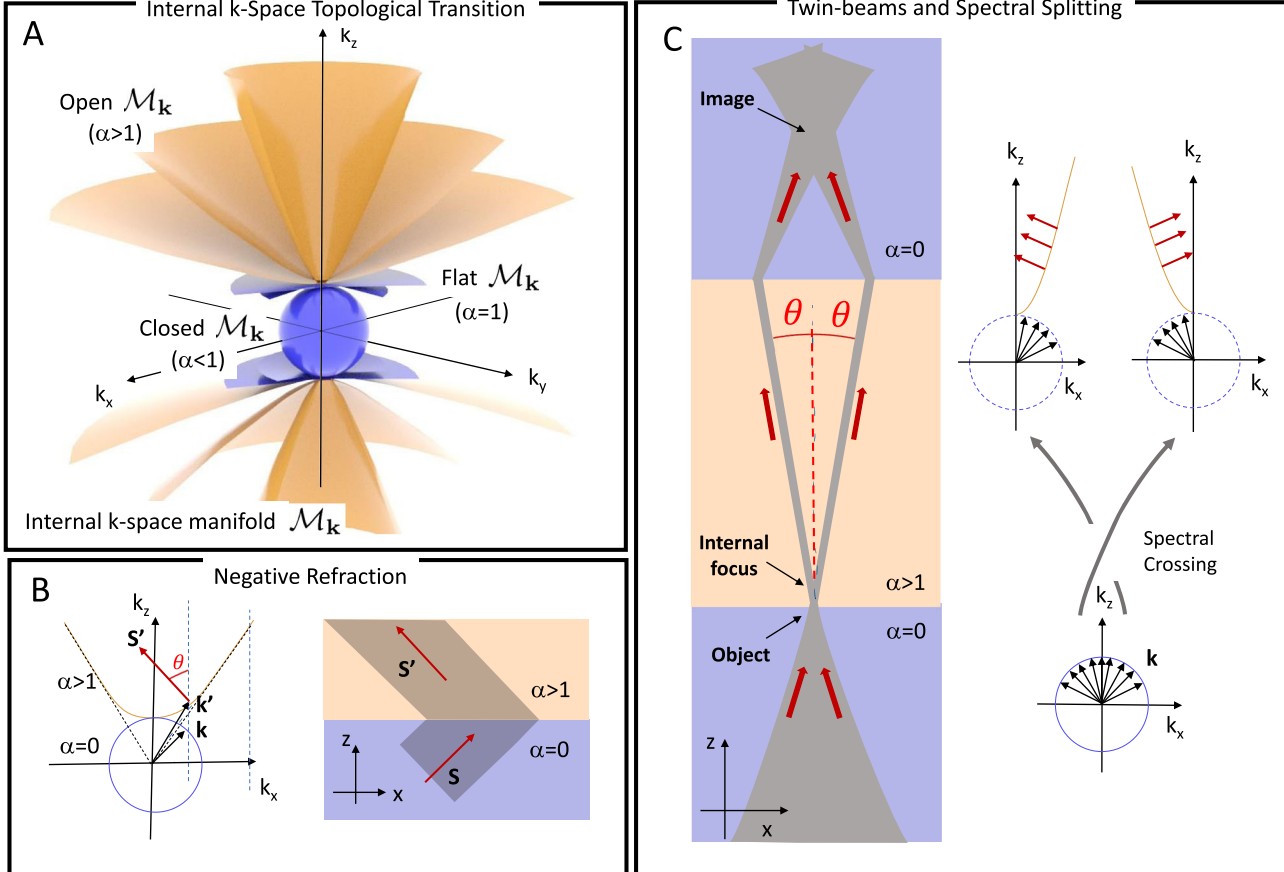

**Fig. 1 A self-induced topological transition. A** Illustration of the internal k-space nonlinear transition as the isofrequency surface $\mathcal{M}_\mathbf{k}$ passes from an closed-surface topology ($\alpha < 1$, blue shade) to an open-surface topology ($\alpha > 1$, yellow shade). **B** Corresponding negative refraction illustrated in the 1+1D case. **C** Illustration of the double k-space transition in the 1+1D case for light passing through a hyperbolic medium ($\alpha > 1$).

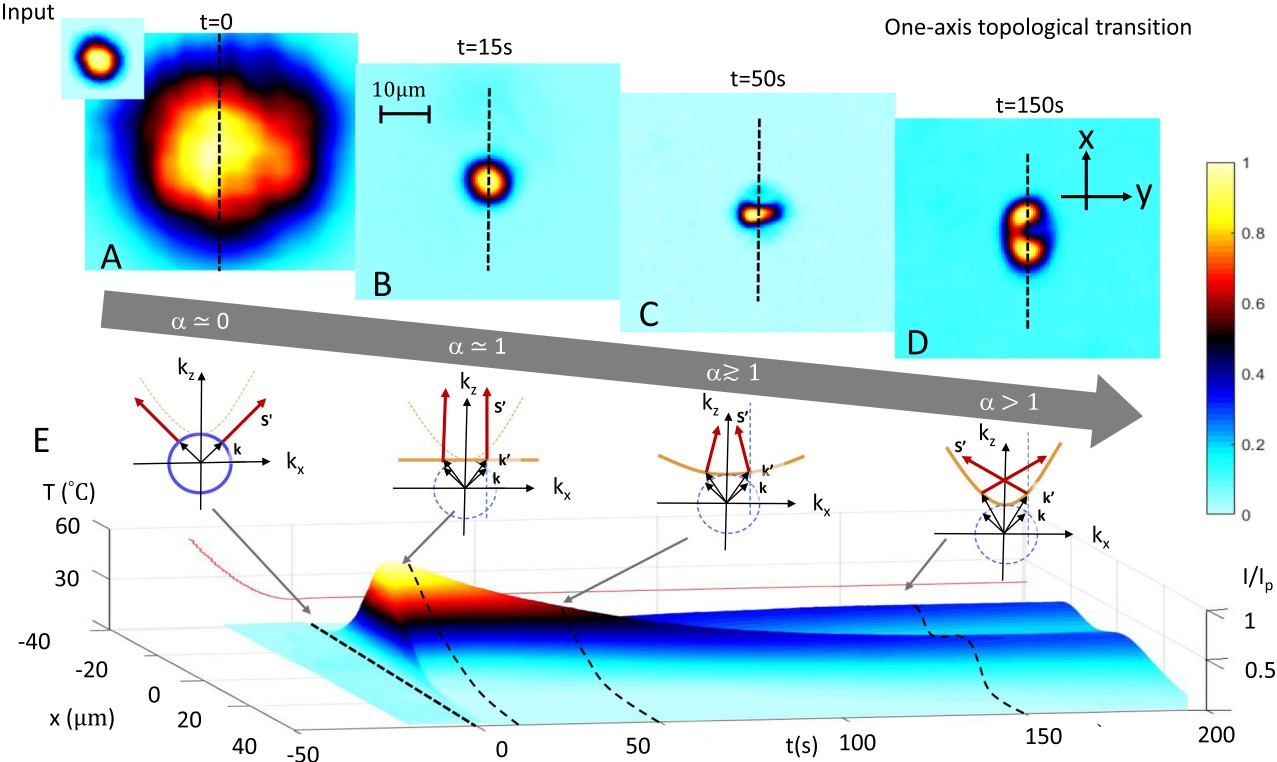

**Fig. 2 Observation of twin-beam formation in a 1+1D topological transition. A** Transverse $xy$ intensity distribution $I$ normalized to the peak intensity of the beam $I_p$ at the input facet as transmitted by the mask (inset) that diffracts to the output after $L_z = 2.2$ mm propagation to 72 µm (FWHM) for $t = 0$ (i.e., $\alpha = 0$). **B** Output distribution for the scale-free optics regime ($t = 15$ s and corresponding $\alpha = 1$). **C** The beam now self-induces the transition at $t = 50$ s, leading to **D** the signature non-diffracting splitting regime for $t > 150$ s ($\alpha > 1$). **E** The self-induced transition seen in time (and corresponding estimated value of $\alpha$) using the $x$ intensity profile (dashed lines in **A–D**) (Source data are provided as a Source Data file).

will once again propagate as the initial localized beam but with a signature non-trivial topological twist that leads to an ensuing crossing point (the 'image').

We here demonstrate experimentally self-induced topological transitions from elliptical to hyperbolic k-space manifolds in room-temperature photorefractive KTN.

## Results

To observe the transition, we carried out experiments using the setup illustrated in Supplementary Fig. 1A. A $\lambda = 488$ nm y-polarized laser is made to propagate (along the $z$ axis) through a potassium-tantalate-niobate (KTN) crystal whose electro-optic response is enhanced through rapid-cooling (see Material section in Methods). A 100-nm-thick aluminum lithographic mask was deposited onto the polished input $xy$ facet, with an etched circular 8 µm diameter transmission pinhole (see the Scanning-Electron-Microscope (SEM) image in Supplementary Fig. 1B. We observed analogous effects for circular pinholes with diameters of 4, 6, and 10 µm (not reported here). The input localized spot ('object') is achieved launching light through the pinhole using a NA = 0.2 lens that forms $a \sim 135$ µm spot centered on it. The spatial beam intensity distribution at the input and output facets of the crystal are monitored using a CMOS detector array and x50 objective lens (NA = 0.4), while beam power is controlled using neutral density filters at input and measured at output using a silicon power-meter.

The photorefractive effect is caused by the electro-optic response to a photoinduced space-charge electric field. The phenomenon is cumulative in the sample exposure time $t$, and $\alpha = \alpha(t)$ reaches a steady-state $\alpha(\infty) = L/\lambda$ only after a transient build-up. For $\alpha(\infty) = \frac{L}{\lambda} > 1$, the topological transition that

occurs as $\alpha$ passes from $\alpha < 1$ to $\alpha > 1$ can be observed as a function of $t$ (see the Controlling $\alpha$ Section in Supplementary Information). Evidence of a self-induced transition is reported in Fig. 2A–D. As $\alpha$ is increased through the transition at $\alpha = 1$, the propagating beam passes from being a conventional diffracting beam dominated by elliptical closed-surface topology (Fig. 2A), to one undergoing the so-called scale-free optics regime associated to the hybrid flat topology (Fig. 2B, $\alpha = 1$). In turn, for higher values of $\alpha > 1$, we find the signature formation of a pair of non-diffracting negative refracting beams, the product of the hyperbolic k-space topology (Fig. 2C, D). In Fig. 2E the full one-axis transition is reported as a function of exposure time $t$.

In Fig. 3 we report the exploration of the transition as the exposure time $t$ is increased and the input launch power $P$ (as measured after passing through the pinhole) is modified. As expected, an increasing value of $\alpha$ leads to an increase in the angle between the $x-$directed twin beams. The increase also leads to a characteristic splitting in the second transverse direction $y$, a phenomenon that becomes ever more evident for higher values of $\alpha$ (third and fourth row of Fig. 3A). For $t = 600$ s, a rhombus-like non-diffracting distribution emerges (fourth row). While splitting in both the $x$ and $y$ directions is expected, as the wave-system is 2+1D, the fact that the splitting first occurs in the $x$ direction and then in the $y$ indicates an anisotropy in the photorefractive response associated to a thermal gradient on cooldown, as observed in previous experiments[34]. As discussed in the Anisotropy Section of Methods, the system obeys an anisotropic $(1 - \alpha_x^2)k_x^2 + (1 - \alpha_y^2)k_y^2 + k_z^2 = k_0^2 n^2$ internal dispersion relationship ($\alpha_y \neq \alpha_x$). Hence, the transition will occur first for one direction (the $x$ direction when $\alpha_x > 1$ while $\alpha_y < 1$) and, only after a sufficient nonlinear build-up, also in the orthogonal $y$ direction (when also $\alpha_y > 1$). The result is a

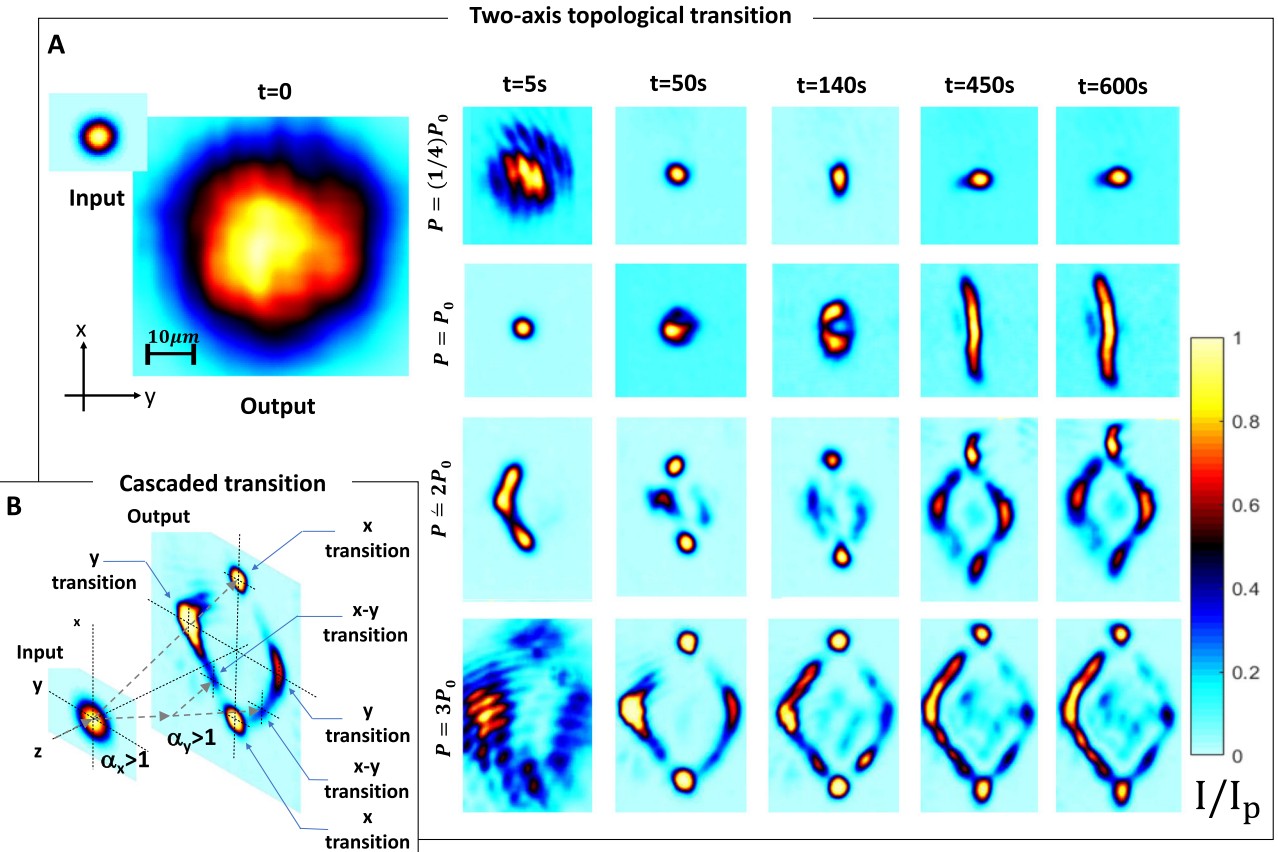

**Fig. 3 Topological transitions in full 2+1D. A** Output intensity distribution for different exposure times and different relative beam power $P/P_0$ (reference $P_0 = 6.5\mu W$ is the power used in the transition reported in Fig. 2). For $P/P_0 = 0.25$ (top row), the input beam (inset left panel) initially diffracts (left panel) after the $L_z = 2.2$ mm propagation and reaches a scale-free optics regime at $\alpha \simeq 1$ at $t = 50$ s, while no signature beam splitting is observed for the reported interval (from $t = 0$ to $t = 600$s). For the same interval, as $P/P_0$ is increased, a signature splitting is also observed along the $x$ and then $y$ direction, leading to a rhombus-like non-diffracting distribution. **B** Results are compatible with the appearance of $x$- and $y$-directed topological transitions, with $\alpha_x \neq \alpha_y$, while the rhombus-like structure signals cascaded transitions dominated by topological protection (see text).

characteristic formation of four non-diffracting beams, the $x$-axis pair separating at an angle such that $\tan \theta_x \simeq \sqrt{\alpha_x^2 - 1}$, while the $y$-axis pair separating at a generally smaller angle $\tan \theta_y \simeq \sqrt{\alpha_y^2 - 1}$. The rhombus-like emission, in turn, signals the occurence of cascaded $x - y$ transitions (see Fig. 3B). Specifically, in distinction to fabricated hyperbolic materials, a self-induced topology acts continuously on the propagating beams, so that a transition can occur also far from the input facet, as determined by the actual intensity at a given position $z$ and the overall exposure at that point $t$. The effect is naturally strongest at the input of the sample ($z \simeq 0$), since here the intensity of the originally diffracting beam is most intense. However, after each of the four negatively refracting beams have formed, amounting to the vertices of the rhombus-like pattern, each will itself seed further cascaded topological transitions, albeit produced by a lower peak intensity and ensuing smaller effective $\alpha$ and corresponding angle $\theta$ (Fig. 3B). Since cascaded transitions in the $x$ direction are protected once a first $x - $axis twin beam transition has occurred, a second cascaded transition can still occur in the orthogonal $y$ direction, as along this direction each of the two $x - $directed beams has its full $k_y$ spectrum. Similarly, $y$-directed beams can suffer a cascaded transition and split in the $x$-direction. In turn, the cascade of the first and second orthogonal topological transitions is, as a whole, topologically protected, since each of the four resulting non-diffracting beams only has a quarter-slice of the original spectrum, a picture

that leads to the signature rhombus-like emission. Interestingly, since photoexcitation from donor impurities is negligible for near IR wavelengths in the Cu-doped samples, the self-induced visible topological transition can even be passively inspected using a second IR probe beam (see Supplementary Information).

The full superlensing effect associated to the 2+1D self-induced transition is reported in Fig. 4 (see Superlens section in Methods). Placing a KTN sample in the path of a diffracting beam from an 'object' (a focused laser beam, Fig. 4A) entails two separate variations in propagation regimes: one entering the material (Fig. 4B) and propagating in a condition of $\alpha > 1$ (Fig. 4C), and a second when exiting the material and returning to the conventional $\alpha = 0$ propagation (Fig. 4D–G). Light, having suffered the double topological transition, now crosses at an external second point, forming a signature 'image' (Fig. 4F).

Importantly, superlensing occurs as a consequence of negative refraction in a solely dielectric medium, without the constraints associated to strong absorption. In this, our study sides recent advances in ultra-thin non-metallic metamaterials for light control[40]. The absence of a metallic component at once guarantees that no wave amplification is required and that no surface plasmons or surface currents are involved so that, combined with the leading linearity in the resulting hyperbolic regime, the standard boundary conditions typical of the Snell law hold. Based on a leading hyperbolic propagation, our scheme involves the passage through the 'inner focus' causing the negative refracting waves to have an

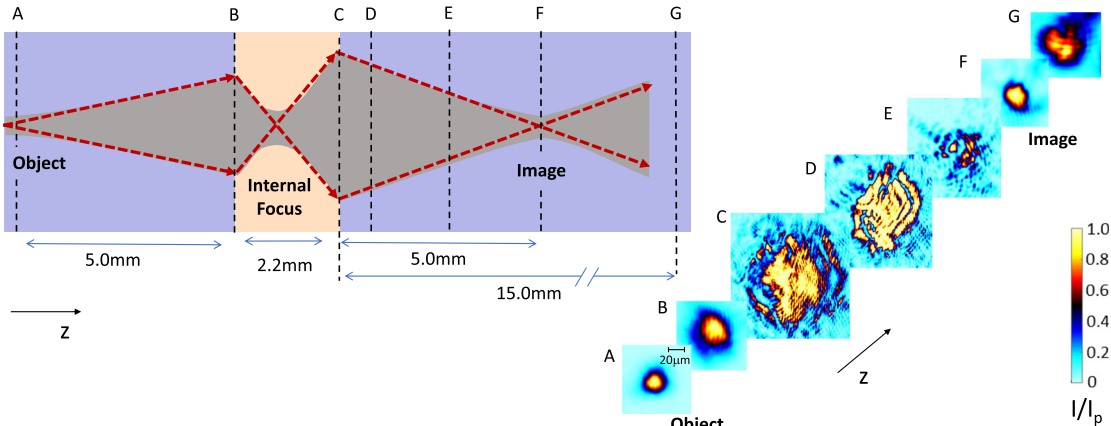

**Fig. 4 Superlensing through a KTN sample. A** Intensity distribution of the beam at its minimum width (`object', 20 µm FWHM) 5 mm from the sample facet, **B** distribution at sample input facet, **C** at the sample output plane after the topological transition, 2.2 mm from the input facet, **D** 0.5 mm from the output facet, **E** 2.5 mm from the output facet, **F** 5 mm from the output facet (`image', 20 µm FWHM), and **G** 10 mm from the output facet.

enlarged (and topologically twisted) output Fourier spectrum, remaining in an overall flat geometry. In these terms, light transmitted through the pin-hole mask, deposited directly on the surface of the sample, will transmit subwavelength features that can be detected, at the output facet, using super-resolution detection schemes. Ideally, a functioning superlens based on a self-induced transition would require a flourescent component in the KTN, as this would convert the high-resolution information directly into propagating waves beyond the sample. Alternatively, the input or output facet of the sample could be sculpted into a curved geometry, as done in a hyperlens[41]. In these conditions, the spatial resolution limit will be dictated by the validity of the macroscopic diffusion-driven model, that breaks down on spatial scales comparable to the average size of the underlying PNRs. Depending on the specific quenching implemented, this places the limit on the scale of tens to hundreds of nanometers[35].

From the applicative perspective, the fact that self-induced hyperbolic dispersion does not involve resonances indicates a method to design broadband flat lenses for visible light. Furthermore, the altered dispersion is spatially resolved, suggesting that it can be used to achieve different transfer functions in parallel across the entire sample.

Conventional nonlinearities, such as the optical Kerr effect, are intensity-dependent, the result being a change in the index of refraction localized in actual space, not in k-space. This can create new topological states, such as self-induced edge states and gap-solitons[42–45]. These arise as topological defects and not as changes in the topology of the plane-wave manifold $\mathcal{M}_k$. The diffusive nonlinearity, in turn, entails the passage from one linear plane-wave regime characterized by the topology of $\mathcal{M}_k$ to another, still linear, regime, with a $\mathcal{M}'_k$ characterized by a different topology. The nonlinear signature of the diffusive nonlinearity remains in the fact that, in distinction to fabricated hyperbolic metamaterials, the modified $\mathcal{M}'_k$ is not the dispersion relation of the KTN crystal, but that of an internal decomposition of the field, a field that is forcibly tied to a localized Gaussian-like waveform. The associated topology of $\mathcal{M}'_k$, therefore, represents an optical analog of the local gauge structure of the field, and not a property of freely propagating plane-waves.

## Methods

**k-space topology**. Optical waves of wavelength $\lambda$ in an isotropic dielectric of index of refraction $n$ are bound to a closed-surface $\mathcal{M}_k$, the Ewald sphere, with $k_x^2 + k_y^2 + k_z^2 = k_0^2 n^2$ and $k_0 = 2\pi/\lambda$. For a given propagation axis, say the z-axis, only the subset of possible $(k_x, k_y)$ eigenpairs such that $k_x^2 + k_y^2 \leq k_0^2 n^2$ leads to propagating

waves with a corresponding real $k_z = \pm\sqrt{k_0^2 n^2 - k_x^2 - k_y^2}$, a constraint that limits the minimum transmitted spatial detail to $\Lambda \sim \lambda/2n$. The curvature of the spherical $\mathcal{M}_k$ leads to standard diffraction for wavepackets, while the Poynting vector **S**, normal to $\mathcal{M}_k$, is parallel to the average **k**. In a metallic-like hyperbolic metamaterial, $\mathcal{M}_k$ is a one-sheet hyperboloid, topologically equivalent to $k_x^2 + k_y^2 - k_z^2 = k_0^2 n^2$, and only the high-resolution spectrum $(k_x, k_y)$ such that $k_x^2 + k_y^2 \leq k_0^2 n^2$ forms propagating waves along z. In a dielectric-like hyperbolic metamaterial, $\mathcal{M}_k$ is topologically equivalent to a two-sheet hyperboloid, such as $-k_x^2 - k_y^2 + k_z^2 = k_0^2 n^2$, and, as long as the macroscopic effective description holds, all values of $(k_x, k_y)$ form propagating waves, no diffraction limit exists, and a constraint holds on the resulting allowed direction of **k** imposed by the minimum value of allowed $k_z^2 \leq k_0^2 n^2$. For both dielectric and metallic hyperbolic systems, spectral components that correspond to an asymptotic conical $\mathcal{M}_k$ $(k_x^2 + k_y^2 \gg k_0^2 n^2)$, lead to negligible diffraction and an **S**, normal to $\mathcal{M}_k$, that is forced along specific topologically protected directions. The connection between hyperbolic dispersion and negative refraction is discussed further in the Supplementary Information.

**Self-induced topology**. In unbiased paraelectric photorefractive crystals, the thermal agitation of photogenerated charge carriers leads to a shape-dependent diffusive nonlinearity[34,35,46–49]. For a monochormatic Gaussian-like optical field **E** of wavelength $\lambda$, light passes from obeying the standard Helmholtz Equation $(\partial_{zz} + \nabla_\perp^2 + k_0^2 n^2)\mathbf{E} = 0$, to a modified but still linear leading equation $(\partial_{zz} + (1 - \alpha^2)\nabla_\perp^2 + k_0^2 n^2)\mathbf{E} = 0$, where $\alpha = L/\lambda$ and $L = 4\pi n^2 \varepsilon_0 \sqrt{g}\chi_{PNR}(K_B T/q)$ at steady-state (see Methods section in DelRe et al.[35]). Here $g$ is the effective quadratic electro-optic coefficient, $\chi_{PNR}$ the effective low-frequency susceptibility of the PNRs, $K_B$ the Boltzmann constant, $T$ the temperature, and $q$ the elementary charge. In terms of plane-wave components $\mathbf{E_k} = \mathbf{E_0}(\mathbf{k}) \exp(i\mathbf{k} \cdot \mathbf{r})$, $\mathcal{M}_k$ is $k_z^2 + (1 - \alpha^2)(k_x^2 + k_y^2) - k_0^2 n^2 = 0$. For conditions in which $\alpha \simeq 1$, beam dynamics are governed by the so-called scale-free-optics equation $(\partial_{zz} + k_0^2 n^2)\mathbf{E} = 0$, where the transverse Laplacian is absent. The result, that is equivalent to the flat-band topology in a metal Lifshitz transition, is the propagation of Gaussian beams of arbitrary intensity and width that simply do not diffract, even when they are initially localized into subwavelength spots[35]. $\alpha > 1$, in turn, corresponds to a self-induced hyperbolic regime in which the Laplacian along the propagation z axis and the transverse x, y axes has opposite signs. Values of $\alpha > 1$ require giant values of $\chi_{PNR} > 10^5$ reached by rapidly quenching the sample to $T_C$[34–36]. The role of nonlocal nonlinearity in the topological transition both in direct and k-space is further discussed in the Supplementary Information.

**Material**. The crystal was grown using the top seeded solution method. A $4.9^{(x)} \times 12.3^{(y)} \times 2.2^{(z)}$ mm sample was cut from the grown boule along the crystallographic [001] axes, and polished to optical grade[50]. The crystal composition is $KTa_{0.65}Nb_{0.35}O_3$, as determined by electron microprobe analysis. The sample was doped, during growth, with traces of Li (less than one substitution of K in 1000) and Cu. Li enhances optical quality close to the phase transition temperature, while Cu supports photorefraction[51–53]. The ratio between Nb and Ta in the solid-solution fixes the value of the Curie Temperature[54], that in the sample is $T_C = 18.4\ °C$, as determined from the maximum point of the relative dielectric permittivity $\varepsilon_r(T)$ curve. The sample is installed in a specifically tailored crystal-holder that is affixed on top of a temperature controlled thermoelectric element (see Supplementary Fig. 1B), while the temperature control circuit is designed to

operate with reduced ringing. To prevent water vapor condensation, the entire optical setup is placed in a box in which a constant flow of dry $N_2$ gas is maintained by overpressure. Prior to each experiment, the crystal is cleansed of residual photorefractive space-charge by illuminating it from the top facet with a laser beam ($\lambda = 532$ nm, diameter of ~4.5 mm, 600 mW for 400 s at a temperature of 80 °C). After this exposure, it is left to dwell for 200 s. From the initial temperature $T(0) = 80$ °C the sample is cooled at a constant cooling rate, ranging between $\dot{T} = 0.3 - 0.8$ °C/s, to the selected dwell temperature $T(\infty) = 16$ °C. When the temperature reaches $T(\infty) + 8$ °C, the controller slows the cool-down, critically relaxing the sample to $T(\infty)$, where it remains unchanged for the duration of the experiment (exhibiting fluctuations of 0.01 °C). For a fixed $T(\infty)$ in proximity of $T_C$, the actual value of $L$ is determined by $\chi_{PNR}$, in turn determined by $\dot{T}$. For KTN, a model picture suggests that for $T > T_C$ the Nb ions emerge from the center of inversion of their respective unit cells forming nanodipoles that hop between the minima of their respective potential field, interact, and create PNRs that fluctuate at random, growing in size on approaching $T_C$. Congruently, cooling to $T_C$, the sample dielectric response exhibits the Vogel-Fulcher-Tammann (VFT) behavior, characteristics of soft condensed matter, behaving as a dipolar-glass-forming-liquid[53].

Experiments in samples with slightly different compositions lead to analogous phenomena with modifications in the specific details, such as Curie point and dependence on exposure time for a specific beam intensity (experiments in different samples are not reported here).

**Anisotropy.** A first basic anisotropic effect is that the topological transition (as reported in Fig. 2) can be seen for an $x$-polarized beam, while a $y$-polarized beam leads only to a weak reduction in diffraction compatible with a low value of associated $\alpha < 1$. This is in agreement with previous studies for which the $\chi_{PNR}$ is found to be small for a polarization parallel to the cooldown thermal gradient and an ensuing pyroelectric field (which is, in our case, principally in the $y$ direction) (see Eqs. (29), (30) in the Supplementary Information of DelRe et al.[34]). The anisotropic dispersion relation recalled in the main article is, in turn, associated to a partial ordering of the PNRs associated to the thermal gradient and the ensuing anisotropic electro-optic response, as discussed in the Supplementary Information.

**Superlens.** The superlens experiment was carried out using a linear polarized $z$ propagating Gaussian beam (488 nm, 46 μW power) focused to its minimum 20 μm FWHM waist in a plane 5 mm before the crystal input facet ('object' plane, Fig. 4A). In other aspects, the system is the same as that reported in Fig. 2, the sample being of the same composition but now with no deposited mask. The beam size at the input facet is a slightly diffracted Gaussian beam with a 50 μm FWHM (Fig. 4B) that, without a topological transition ($T > T_C$), diffracts to an 80 μm FWHM in a plane 5 mm from the output facet ('image' plane). Results are reported for $t = 100$ s, while the sample is rapidly cooled from $T(0) = T_C + 64$ K to $T(\infty) = T_C - 3$ K (in this specific case, $\dot{T} \simeq 0.5$ °C/s). As expected for the resulting superlens effect, the beam expands as it reaches the output facet (Fig. 4C), after the internal focus point, and refocuses on exiting the crystal (Fig. 4D–F) to then re-expand after the image focus (Fig. 4G). The position of the focus, that in the reported case is 5 mm from the output facet, is dependent on the cooling rate $\dot{T}$ and exposure $t$, and was observed in the range of 2–5 mm from the output facet changing $\dot{T}$ in the range $0.3 - 0.8$ °C/s. It is useful to discuss the details in terms of the transverse spatial spectrum of the superlensing in Fig. 4 compared to the experiments in Figs. 2 and 3. Experiments in Figs. 2 and 3 involve a transverse spectrum with Airy rings and a characteristic $|\mathbf{k}_\perp| \sim 5 \times 10^5$ m$^{-1}$ able to populate, for the values of $\alpha$ inspected, regions of the hyperbolic branches. In turn, in the conditions of Fig. 4, the spectrum is a localized Gaussian ($|\mathbf{k}_\perp| \sim 5 \times 10^4$ m$^{-1}$), and the propagation is dominated by the central parabolic region of the dispersion hyperbola.

## Data availability
The datasets analyzed in the current study are available from the corresponding author on reasonable request. Source data are provided with this paper.

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

## Acknowledgements

Support from the Sapienza-Ricerca di Ateneo 2019 and 2020 projects (C.C. and E.D.R.), the H2020 Fet project PhoQus (C.C. and E.D.R.), the PRIN 2017 PELM (grant number 20177PSCKT, C.C. and E.D.R.) and PRIN 2020 (grant number 2020X4T57A, E.D.R.) projects, the Israel Science Foundation (Grant No. 1960/16, A.J.A.), and the Israel ministry of Science technology and space (Grant No. 3-16816, A.J.A.) is acknowledged.

## Author contributions

A.J.A. designed the crystals, that were grown and synthesized by Y.Ga. and G.P.; A.J.A. conceived the propagation experiments, while Y.Ge. and E.D.R. conceived the super-lensing experiments. Y.Ge. carried out the principal measurements under the supervision of A.J.A.; E.D.R. elaborated, with the help of Y.Ge., the physical framework and interpretation, discussing ideas with F.D.M., C.C., A.J.A . F.D.M., S.F., and L.F. carried out preliminary experiments. All authors participated in discussions. Y.Ge. and E.D.R. wrote the article, with the help of all authors.

## Competing interests

The authors declare no competing interests.
