## [Peer Review File · Nature Communications]

Hyperbolic optics and superlensing in room-temperature KTN from self-induced k-space topological transitionsReviewers' Comments:

Reviewer #1:

Remarks to the Author:

This paper claims the experimental observation of a transformation of "a bulk KTN perovskite into a broadband 3D hyperbolic substance for visible light" that is attributed to the action of nonlinearity that can transform normal refraction into negative refraction due to giant electro-optic response to the electric field.

This paper, while presenting some valid experimental results, is misleading in general, and it is based on wrong assumptions and interpretations.

First, a hyperbolic medium is a medium with a hyperbolic dispersion, and it does not necessarily support negative refraction and superlensing, these effects depend on the averaged refractive indices of the effective media. Second, a wave dispersion is always associated with linear media, and it does not have meaning for pulses and beams in a nonlinear medium. The explanation of the observed effects via a modification of the linear dispersion is incorrect.

In this paper, the authors indeed observe a negative refraction from a homogenous material, but this effect is known [see e.g., Y. Gao et al, Phys. Rev. Lett. 104, 034501 (2010) and later papers that can be found via citations to that paper in Google Scholar], and it is not associated with a hyperbolic media or "phase transitions"

Reviewer #2:

Remarks to the Author:

This manuscript reports theory and experiments on a new class of topological materials, created by nonlinear light-matter interactions in a KTN perovskite crystal.

I find the article well-written, timely, and sound. The topic presented is new and fits well the criteria for publication in NCOMM. As such, I am in favor of the publication of this work. I have some suggestions, mainly to improve the outreach of the paper:

1) As the paper discusses light control with metamaterials in the introduction, it should cite the recent work, which discusses a new approach to create ultra-thin metamaterials for engineering any type of light control:

a) Getman et al. Light: Science & Applications (2021), <https://doi.org/10.1038/s41377-021-00489-7>

2) I find it very hard to read the figure insets: the texts and the images themselves are too small. The authors can consider splitting some of the insets as separate images or using the SI.

3) The derivation of the "modified but still linear leading equation" is not sketched, nor a citation is provided. I would add a citation, or explain briefly how the equation is derived, as this equation is quite important for the paper.

4) The section on controlling alpha, which is an important parameter, is split into two parts, one in the method and (a larger part) in the SI. The authors can consider creating a single section where this is explained, or add the method text in the main text and refer to the rest in the SI. This can improve readability: the manuscript is quite massive in terms of theoretical results and experiments, and having a streamlined narrative can significantly help the reader appreciate the novelty of this interesting work.

Reviewer #3:

Remarks to the Author:

The manuscript shows how nonlinearity can transform a bulk KTN perovskite into a broadband 3D hyperbolic substance

for visible light due to the giant electro-optic response to the electric field generated by the thermal diffusion of photogenerated charges. The negative refraction and superlensing at room-temperature has been demonstrated. These results may open new scenarios in the exploration of enhanced light-matter interaction and in the design of broadband photonic devices. The scheme is interesting, however some issues should be clear:

1. What is the difference between the proposed self-induced topological transition and the self-induced topological effects supported by standard nonlinearities? And then I don't really understand the difference .
2. What is the influence of the nonlinearities on the negative refraction and superlensing? Such as, the intensity, polarization, pulse width, repetition frequency, and etc.
3. In the superlensing, what is the diffraction limit in this nonlinear KTN? Can it break the diffraction limit? Please give the physical interpretation and the possible experimental validation.
4. Please give some practical application scenarios with this self-induced k-space topological transitions.
5. Please give the factors that affect the internal focus.

===== Response to Report of Reviewer 1 =====

===== Report of Reviewer 1

This paper claims the experimental observation of a transformation of “a bulk KTN perovskite into a broadband 3D hyperbolic substance for visible light” that is attributed to the action of nonlinearity that can transform normal refraction into negative refraction due to giant electro-optic response to the electric field.

This paper, while presenting some valid experimental results, is misleading in general, and it is based on wrong assumptions and interpretations.

First, a hyperbolic medium is a medium with a hyperbolic dispersion, and it does not necessary support negative refraction and superlensing, these effects depend on the averaged refractive indices of the effective media.

===== Response

The statement expresses no objection to the content of our manuscript. As stated in an important review article on hyperbolic metamaterials (<https://www.nature.com/articles/nphoton.2013.243>)

"Here, we review hyperbolic metamaterials - one of the most unusual classes of electromagnetic metamaterials. They display hyperbolic (or indefinite) dispersion, which originates from one of the principal components of their electric or magnetic effective tensor having the opposite sign to the other two principal components. Such anisotropic structured materials exhibit distinctive properties, including strong enhancement of spontaneous emission, diverging density of states, negative refraction and enhanced superlensing effects."

I hope we can all agree, in agreement with this article, that a hyperbolic medium has hyperbolic dispersion, the result being that it can manifest superlensing and negative refraction. It goes without saying that, as for any physical statement, a hyperbolic material need not always manifest negative refraction and superlensing. Specifically, negative refraction occurs when passing from a material or regime with elliptical dispersion to one with hyperbolic dispersion (or, indeed, viceversa). Evidently, if such a discontinuity or passage does not occur, neither does negative refraction. The situation is clarified in many articles that address wave propagation in hyperbolic materials. Just to be specific, we can pick the article cited by Referee 1 in the report (Gao et al. PRL 104, 034501 (2010)). Here it is clarified that, for specific polarizations and wavelengths, the dispersion is hyperbolic (see eq.(3) therein) and this is predicted to lead to negative refraction (eq.(4)-eq.(6)). Equally for superlensing. Specifically, superlensing will be evident in a hyperbolic material if i) a sufficient amount of spatial spectrum (transverse spatial k vectors) is excited so as to distinguish between diffraction and antidiffraction for the given propagation length, and ii) a parabolic-like portion of the isofrequency dispersion relation is excited. As is explicitly detailed in our manuscript, Fig.1, 2, and 3 refer to conditions in which negative refraction is dominant, while Fig. 4 refers to experiments where superlensing is dominant.

Furthermore, the report seems to imply the notion that hyperbolic dispersion can somehow not be associated with an average. Dispersion in a material is, by definition, a product of the macroscopic Maxwell Equations and hence is an averaged quantity.

In short, this first point contains no objection to our paper and addresses issues, such as averaging and dispersion, in an incorrect manner.

===== Report of Reviewer 1

Second, a wave dispersion is always associated with linear media, and it does not have meaning for pulses and beams in a nonlinear medium.

=====
Response

This statement is incorrect. A textbook on waves will begin by underlining that a macroscopic wave theory, such as that associated to the macroscopic Maxwell Equations, is only approximately linear (see, for example, *Nonlinear Optics - Boyd*). That wave dispersion has meaning both for linear and nonlinear media is the very basis for all our present and established understanding of nonlinear optics and, indeed, of nonlinear waves in general. How would any of us ever venture to achieve phase-matching, the most basic and general scheme in nonlinear photonics, if -literally- dispersion were not valid for nonlinear media and pulses? Indeed for pulses, linear dispersion is altered by self-phase modulation associated with a small correction to the linear susceptibility (the Kerr effect). Analogously, linear diffraction in self-focusing media is altered by self-lensing. Evidently, wave dispersion and nonlinearity necessarily coexist in a nonlinear medium, the paradigm being a soliton. A soliton in an optical fiber exists exactly because linear dispersion is precisely balanced by self-phase modulation (see, for example, *Optical Solitons – Kivshar, Agrawal*).

Focusing on our specific manuscript, we have amply underlined that waves that obey a specific linear wave equation are described by the isofrequency curve, while waves that obey a nonlinear wave equation are described by the combined effect of the isofrequency curve and the nonlinear correction. In our case of a diffusion-driven nonlinearity, as described in our manuscript and in previous literature (see, for example, *DelRe2011* and *DelRe2015* in the bibliography), the nonlinearity causes the system to be described by a leading linear wave equation that is different from the original Helmholtz Equation. The passage from the original elliptical Helmholtz Equation to the hyperbolic Klein-Gordon-like wave equation constitutes the physical background for the topological transition demonstrated in this present paper.

=====
Report of Reviewer 1

The explanation of the observed effects via a modification of the linear dispersion is incorrect.

=====
Response

No further justification for this conclusion is provided, while both previous arguments, as discussed above, are unmotivated. It follows that this conclusion is not tenable.

The manuscript makes it quite clear that our model explains in detail our observations. While no level of explanatory power can render a model the only explanation to a phenomenon, yet a model capable of describing observations is to be considered a valid model and certainly not an “incorrect” model, as framed in the report. Regarding the details of the model itself, the underlying structure (as described in the Self-induced topology section of the Methods) has been established in a number of papers

(including Crosignani et al., *Opt. Lett.* 23, 912 (1998), Crosignani et al., *Phys. Rev. Lett.* 82, 1664 (1999), Conti et al., *Phys. Rev. A* 84, 043809 (2011), Di Mei et al., *Opt. Exp.* 22, 31434 (2014), DelRe et al., *Nat. Photon.* 5, 39-42 (2011), DelRe et al., *Nat. Photon.* 9, 228-232 (2015), all cited in the manuscript]) that follow and thread into other k-space linear and nonlinear explorations (including Firstenberg et al., *Nat. Phys.* 5, 665-668 (2009), Kosaka et al., *Appl. Phys. Lett.* 74, 1212-1214 (1999), Eisenberg et al. *Phys. Rev. Lett.* 85, 1863-1866 (2000), cited in the manuscript). Its linearization, described here, shows how the effect is

actually an internal (gauge) version of the well-established model describing hyperbolic metamaterials (see, for example, Poddubny et al., Nat. Photon. 7, 948-957 (2013), cited).

=====
Report of Reviewer 1

In this paper, the authors indeed observe a negative refraction from a homogenous material, but this effect is known [see e.g., Y. Gao et al, Phys. Rev. Lett. 104, 034501 (2010) and later papers that can be found via citations to that paper in Google Scholar], and it is not associated with a hyperbolic media or “phase transitions”.

=====
Response

This statement is misplaced. The report of Referee 1 misrepresents the interesting numerical work of Gao 2010 as described by those authors.

First of all, the exact opposite of what claimed in the Report is stated in the cited paper (Y. Gao et al, Phys. Rev. Lett. 104, 034501 (2010)), and indeed later papers that cite it: results addressed therein are explicitly associated to hyperbolic media and “phase transitions” from abstract down into the details of the main text.

Just to put it on record, this is what the abstract of the Gao2010 paper states:

“We numerically demonstrate optical negative refraction in ferrofluids containing isotropic Fe₃O₄ nanoparticles, each having an isotropic Ag shell, in the presence of an external dc magnetic field H. The all-angle broadband optical negative refraction with magnetocontrollability arises from H-induced chains or columns. They result in hyperbolic equifrequency contour for transverse magnetic waves propagating in the system. The finite element simulations verify the analyses using the effective medium approximation. Experimental demonstration and potential applications are suggested and discussed”

Reference to a “hyperbolic equifrequency contour”, the signature of a hyperbolic material as described in literature and in our manuscript, is quite explicit. In the main text of that article we find the entire hyperbolic description in Eq.(1), (2), and finally Eq.(3). There is very little more that can be said. Analogously for “phase transitions”: the cited article is about how an external magnetic field can induce a ferrofluid to manifest a columnar phase. For example, we read

“For ferrofluids, the field-induced columnar phase with equal spacing was experimentally reported [11,12] in confined ferrofluid films subjected to an in-plane H, and the dipolar interaction between the chains (which are locally displaced in a hexagonal fashion) gives rise to their arrangement in columns.”

But even all this said, we did not observe negative refraction from a homogeneous material, as indicated by Referee 1. As described in our manuscript, we observed negative refraction in a bulk solid-state crystal on consequence of a photoexcited charge diffusion-driven nonlinearity. It is evident that a nonlinear material cannot be considered simply a homogeneous material (see the detailed description of the underlying nonlinear response associated to polar-nanoregions in DelRe *et al.*, Nat. Photon. 5, 39-42 (2011), supplementary information). In fact, we used the word “homogeneous” only in describing the material when an IR beam propagates in conditions where no space-charge is generated and, congruently, no hyperbolic dispersion is found (see supplementary information of the present article).

Summing up, the contents of the Report of Referee 1 are unsound, the statements are misplaced, and the discussion is erroneous. As detailed above, in distinction to what asserted in the Report, the assumptions and interpretation provided in our manuscript are

- i) based on well-established concepts and models that are rooted in previous literature,
- ii) provide an adequate description of the phenomena observed,

iii) provide the basis for further developments, possible applications, and experiments.

=====
===== Response to Report of Reviewer 2 =====

=====
===== Report of Reviewer 2

This manuscript reports theory and experiments on a new class of topological materials, created by nonlinear light-matter interactions in a KTN perovskite crystal.

I find the article well-written, timely, and sound. The topic presented is new and fits well the criteria for publication in NCOMM. As such, I am in favor of the publication of this work.

=====
===== Response

We thank the Reviewer for having analyzed our paper and for the positive stance.

=====
===== Report of Reviewer 2

I have some suggestions, mainly to improve the outreach of the paper:

1) As the paper discusses light control with metamaterials in the introduction, it should cite the recent work, which discusses a new approach to create ultra-thin metamaterials for engineering any type of light control:

a) Getman et al. Light: Science & Applications (2021), <https://doi.org/10.1038/s41377-021-00489-7>

=====
===== Response

Yes, thank you for the suggestion, we have added the reference as per Change 1.

=====
===== Report of Reviewer 2

2) I find it very hard to read the figure insets: the texts and the images themselves are too small. The authors can consider splitting some of the insets as separate images or using the SI.

=====
===== Response

Yes, agreed. We have fixed the problem as per Change 2.

=====
===== Report of Reviewer 2

3) The derivation of the "modified but still linear leading equation" is not sketched, nor a citation is provided. I would add a citation, or explain briefly how the equation is derived, as this equation is quite important for the paper.

=====
===== Response

Yes, thank you, we have fixed the problem, as per Change 3.

=====
===== Report of Reviewer 2

4) The section on controlling alpha, which is an important parameter, is split into two parts, one in the method and (a larger part) in the SI. The authors can consider creating a single section where this is explained, or add the method text in the main text and refer to the rest in the SI. This can improve readability: the manuscript is quite massive in terms of theoretical results and experiments, and having a streamlined narrative can significantly help the reader appreciate the novelty of this interesting work.

=====
===== Response

Yes, we have acted appropriately combining the parts together in a single section, as per Change 4.

===== Response to Report of Reviewer 3 =====

===== Report of Reviewer 3

The manuscript shows how nonlinearity can transform a bulk KTN perovskite into a broadband 3D hyperbolic substance for visible light due to the giant electro-optic response to the electric field generated by the thermal diffusion of photogenerated charges. The negative refraction and superlensing at room-temperature has been demonstrated.

These results may open new scenarios in the exploration of enhanced light-matter interaction and in the design of broadband photonic devices.

The scheme is interesting, however some issues should be clear:

1. What is the difference between the proposed self-induced topological transition and the self-induced topological effects supported by standard nonlinearities? And then I don't really understand the difference.

===== Response

We thank Reviewer 3 for having analyzed our manuscript and for his overall positive stance.

Answer to Point 1. We agree with the Referee that topology plays a fundamental role also for nonlinear waves and solitons. In turn, the phenomenon we are targeting here is specifically a transition that occurs when the k-space manifold of degenerate linear plane-wave solutions goes from one topology to another. The idea of this topological transition was originally introduced to describe possible effects associated to the behavior Bloch-electrons in a conductor close to zero T and subject to pressure (Lifshitz1960). The interesting effects could then be described by the change in the topology of the Fermi surface that could pass, under pressure, from an ellipsoid to a one or two-sheet hyperboloid. In an optical system, the topological transition becomes the transition that occurs for propagating waves when the standard k-space Ewald sphere or ellipsoid that governs the isofrequency propagation associated to the Helmholtz Equation is transformed, in response to some external stimuli, into a topologically different k-space surface, such as a one or two-sheet hyperboloid. The point is that the transition is occurring in k-space, not in actual space. Conventional nonlinearities, such as the Kerr effect, are associated to the local intensity of light in direct space. This leads, in general, to wave-mixing and self-phase modulation, effects that produce localized waves in direct space that are not solely the result of the isofrequency surface. So, for example, nonlinearity can produce topological gap solitons and edge states: these are localized waves that violate the basic wave phenomenology that is associated to wave dispersion and dynamics as deduced directly and solely from the topology of the k-surface (Smirnova et. al., *Appl. Phys. Rev.* 7, 021306 (2020), Hadad et al., *Nat. Electron.* 1, 178-182 (2018), cited). In other words, in terms of the isofrequency surface, edge states and gap solitons are equivalent to rips and defects in the topology. In fact, some solitons (topological solitons) behave in manner that is very similar to topological defects, localized regions in actual space where a broken symmetry manifests a singularity, such as a domain wall or a domain vortex.

The nonlinearity discussed in the present manuscript, on the other hand, acts to change the shape and topology of the effective k-space manifold, that is, it acts in k-space, not in direct space. It is this that makes the self-induced topological transition in k-space possible. As detailed in the manuscript, indeed this is not true literally for plane wave solutions, but for the leading description of Gaussian-like beams. In this sense, the shape-dependent nonlinearity acts to change the topology of an internal symmetry, the point dependent generalization of the standard global symmetry that is associated to a fabricated isofrequency dispersion.

To further underline this point, we have added extra explanatory statements, as per Change 5.

===== Report of Reviewer 3

2. What is the influence of the nonlinearities on the negative refraction and superlensing? Such as, the intensity, polarization, pulse width, repetition frequency, and etc.

===== Response

As described in the Superlens section in Methods , negative refraction and superlensing are governed by α . This fixes the angle of negative refraction through the relationships described in the first paragraph of the main (for the asymptotic branches), the ensuing position of the inner and of the external second focus, as also determined by the length of the sample L_z . α also fixes the overall shape of the hyperbola, determining the regions of the transverse spatial spectrum that suffer negative non-diffracting angled propagation (leading to the characteristic twin-beam propagation) and those that suffer a parabolic-like superlensing.

The value of α depends on the exposure time and intensity distribution through the cumulative relationship described in the Self-Induced Topology section of Methods .

The value of α also depends directly on the PNR dielectric susceptibility, that depends on the beam peak intensity and cooling rate, as detailed in the supplementary information.

Since the response is cumulative, pulse width and repetition rate enter the picture only in determining the average peak intensity (in space), as long as the overall effective response time is still much longer than the pulse width.

A key parameter for a given value of α is then the actual beam spatial transverse spectrum, that depends on the size and specific shape of the input beam. In order to clarify the specific role of α in fixing the details of the optical propagation, we have added a series of clarifications, as per Change 6.

===== Report of Reviewer 3

3. In the superlensing, what is the diffraction limit in this nonlinear KTN? Can it break the diffraction limit? Please give the physical interpretation and the possible experimental validation.

===== Response

As for metamaterials, so also here we are always operating in the regime of validity of a macroscopic Maxwell Equation regime. It follows that while the diffusion-driven nonlinear wave model has no theoretical diffraction limit, its predictions are only valid on scales in which the nonlinearity is valid, that is, on scales larger than the ferroelectric polar-nanoregions. This has been previously analyzed in Reference (DelRe et al., Nat. Photon. 888, 999 (2015), cited) using a super-resolution structured illumination imaging technique, and suggests that below several tens to hundreds of nanometers (depending on the cooling rate implemented) the direct role of the underlying polar-nanoregions becomes dominant. Put differently, the hyperbolic dispersion has no diffraction limit and no evanescent-wave spectrum, but it is only valid for k -vectors that allow light to average on the PNRs. As regards to experimental validation, this requires the use of super-resolved imaging at the output of the sample. As for conventional hyperbolic materials, superlensing per se does not lead to super-resolution, as light entering the material and leaving it must be propagating in the standard elliptic encasing material, in our case air. So the experiment reported in Fig. 4 demonstrates superlensing, but cannot be considered an experimental validation of super-resolved imaging. In turn, the experiments in Figs.1-3 are associated to a super-resolved field produced by a mask deposited on the sample. In this case, we predict, as for conventional hyperbolic materials, that the inner focus and in general the inner waves carry all the information (the limit being the aforementioned scale of the polar nanoregions), but on exiting the sample, the super-resolved portion of the spectrum remains trapped in the sample, suffering the hyperbolic equivalent of total internal reflection. In fabricated systems,

this limitation has been addressed using a curved material, able to geometrically transform the inner spectrum into propagating outer waves (an elaboration of the hyperlens designs, see, for example, Lu et.al, *Nature Photon.* 3, 1205 (2012), added). Curved samples of KLTN are in our future developments, but we have, to date, no such sample. A different strategy is to use temperature to possibly alter the input and output processes, again, something we plan in the future. We have added specific statements on the matter, as per Change 7.

=====
Report of Reviewer 3

4. Please give some practical application scenarios with this self-induced k-space topological transitions.

=====
Response

One natural application is to have a flat lens system that is non-resonant (broadband). Another possible application is that the effect actually occurs in the bulk of a system locally, so, the hyperbolic dispersion can occur for different multiple beams, even crossing each other, a super-resolved basis for optical interconnects. Furthermore, it would be interesting to harness hyperbolic dispersion in combination with the formation of a ferroelectric supercrystal for electronic nonlinear optical effects, but this is an entirely new realm of study that is, at present, still under study. Finally, the hyperbolic dispersion can also be used to route and affect photorefractively inactive beams, such as the IR case described in the supplementary information. We have added statements, as per Change 8.

=====
Report of Reviewer 3

5. Please give the factors that affect the internal focus.

=====
Response

Yes, the response is given as part of the response to the previous point 2.

Reviewers' Comments:

Reviewer #1:

Remarks to the Author:

The revised version is clearer, and the concept and motivation are more transparent. I support publication

Reviewer #2:

Remarks to the Author:

accept

Reviewer #3:

Remarks to the Author:

I also support the opinion of review 1, a hyperbolic medium is a medium with a hyperbolic dispersion, and it does not necessary support negative refraction and superlensing, the author should give the physical interpretation or fullwave simulations of their viewpoint, rather than cite some references.

2. The explanation of the observed effects via a modification of the linear dispersion is not clear, I still can understand what is the role of the nonlinearity? The authors should give more intuitive explanations or experimental results.

3. Please give the simple explanation what the difference between the results in real space and K-space.

===== Response to Second Report of Reviewer 1 =====

===== Second Report of Reviewer 1

The revised version is clearer, and the concept and motivation are more transparent. I support publication

===== Response

We thank Reviewer 1 for the positive stance.

===== Response to Second Report of Reviewer 2 =====

===== Second Report of Reviewer 2

accept

===== Response

We thank Reviewer 2 for the positive stance.

===== Response to Second Report of Reviewer 3 =====

===== Second Report of Reviewer 3

1. I also support the opinion of review 1, a hyperbolic medium is a medium with a hyperbolic dispersion, and it does not necessary support negative refraction and superlensing, the author should give the physical interpretation or fullwave simulations of their viewpoint, rather than cite some references.

===== Response

To address point 1, we have added a more explicit discussion on the relationship between hyperbolic dispersion, negative refraction, and superlensing in the SI. The paragraph is called out in the main manuscript from the Methods Section, as per Change 1.

===== Second Report of Reviewer 3

2. The explanation of the observed effects via a modification of the linear dispersion is not clear, I still can understand what is the role of the nonlinearity? The authors should give more intuitive explanations or experimental results.

3. Please give the simple explanation what the difference between the results in real space and K-space.

===== Response

To address both points 2 and 3, we have added a more explicit discussion on the role of diffusion, nonlocality, and nonlinearity in the SI, along with a comparison between the interpretation of results in k-space and in direct space. The new material is called out in the main manuscript from the Methods Section, as per Change 2.

Reviewers' Comments:

Reviewer #3:

Remarks to the Author:

The revised version can be accepted in the current version.